# Normative Corporate Income Tax with Rent for SDGs' Funding: Case of the U.S.

Mihoko Shimamoto 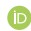

Faculty of Social Sciences, Hosei University, Aihara-cho 4342, Machida-shi, Tokyo 194-0298, Japan;
mihokos@hosei.ac.jp; Tel.: +81-42-783-2393

**Abstract:** The purpose of this study is to explain the justification for taxing corporate rents as a funding source for Sustainable Development Goals (SDGs), and to calculate a normative corporate tax rate that takes into account rents for corporations, especially multinational corporations, and to recommend that the current corporate tax surcharge be used to finance social common capital. Considering global tax avoidance, we propose that many countries cooperate to raise their corporate taxes and finance SDGs. Aiming to calculate a normative corporate tax rate with rents for each country, we applied the total factor productivity method for calculating the markup rate, assumed long-term interest rates to be the marginal efficiency of capital, and developed a normative corporate tax rate calculation method. Using a Cobb–Douglas function in dynamic pseudo-competitive profit optimal conditions, we calculated the rents of 234 American corporations listed on the S&P 500 index. The normative tax rates from 1982 to 2014 for these companies are stable at 40 to 60%, whereas corporate income tax has gradually decreased from 40% to less than 30%. Thus, the amount lost due to the race to the bottom of corporate taxes can be used to finance the SDGs.

**Keywords:** SDGs; social common capital; dynamic rent; normative corporate income tax; international taxation; U.S. companies

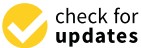



## 1. Introduction

In September 2015, with 17 Sustainable Development Goals (SDGs) at its core, the 2030 Agenda for Sustainable Development was adopted at the UN Sustainable Development Summit. To achieve these 17 SDGs, the improvement and capacity building of environmental and various social services and income redistribution systems are necessary. Uzawa [1] had referred to them as social common capitals. He stated that "It is generally classified into three categories: natural capital, social infrastructure, and institutional capital (p. 3)".

Raising financial resources always presents major difficulties. Although, it is conceivable to use the tax revenue as a double dividend of the environmental tax to control the emission where the cause of the problem is the emission of $CO_2$, the application of environmental taxes is almost exclusively in the area of emissions of certain substances. Moreover, a commodity tax has the weaknesses of regressive taxation. Indeed, amid fears of a global recession and soaring energy prices, the recent introduction or increase in carbon taxes, for example, is facing headwinds. Funding is the biggest point of contention for the Green New Deal, which is often proposed by progressive parties. In 2022, COP27 reached a breakthrough agreement on a new "Loss and Damage" fund for vulnerable countries. Funding for international issues generally seeks to build resources based on monetary contributions from member countries of international treaties or organisations. What kind of revenue source, then, is not regressive and does not drag down the economy? One approach is to tax corporate rents. This is because companies are producing while enjoying the social common capital supplied to them, but it has not seemed that they are being adequately compensated for it. Nevertheless, a uniform increase in corporate taxes could have a negative impact on the economic progress. Therefore, we propose that by taxing

the portion of enjoyed monopsony rents that belong to other agents or nature, we can raise funds for social common capital while minimizing regressivity and the impact on the economy.

The purpose of this study is to explain the justification for taxing corporate rents as a funding source of social common capital procurement, and to calculate a normative corporate tax rate that considers rents for corporations, especially multinational corporations, and to recommend that the corporate tax surcharge be used to finance social common capital.

In Section 1, we analyse the reasons why corporate income tax would be desirable to pay for social common capital, based on the knowledge of environmental economics and public finance. Section 2 describes the normative corporate income tax rate and its calculation method.

In Section 3, we estimate trends in the normative corporate tax rate for 234 large U.S. corporations since the 1980s. Section 4 discusses key findings and policy recommendations.

### 1.1. Why the SDGs Should Be Financed by Corporate Rent Taxation

Why, then, should we connect corporate rent taxation as a source of revenue to achieve SDGs? The first reason is rationality in resource allocation.

### 1.1.1. SDGs and Monopsony and Free-Ride by Firms

The SDGs are largely divided into environmental issues and those about human rights, education, and employment. These distortions in resource allocation can be divided into two parts in economic theory: the issues of monopsony in the input markets of resources and labour for firms and their free-ride for public services. The extraction process of various resources by firms does not often consider the environment and human rights. (According to the approach of ecological economics, environmental issues can be divided into two types: resource management issues that occur at the stage of resource uptake from the ecosystem to the economic sub-system, and emission issues that occur at the stage of emission from the economic sub-system to the ecosystem. Emissions have been addressed through environmental taxes and emissions trading, although resource issues tend to be less visible to consumers in developed countries and policy instruments must be more complex.) Specific issues include mineral and water resource development, tropical forest destruction, and soil salinization owing to unsustainable agriculture. Ultimately, these are incorporated into the production process as cheap raw materials that are advantageous to the companies with price control, while creating such external diseconomies.

Certainly, an increasing number of companies have become more environmentally conscious in recent years. However, international organisations and NGOs have repeatedly reported that it remains insufficient. For example, although more than 100 world leaders have promised to end and reverse deforestation by 2030 in the COP26, the UN Climate Change High-Level Champions [2] stated that "Of the nearly 150 companies considered critical for tackling tropical deforestation that have already committed to net zero, just 9 companies (6%) are making strong progress on deforestation" in its 2022 report on company net-zero targets. Thus, the demand side will continue to enjoy monopsony rents.

In addition to environmental issues, the SDGs include several goals related to employment of workers, education for the development of human capital, improvement of living conditions, and human rights. Another major input in the production process for producers is labour. Low wages, instability of employment status, and poor welfare benefits for workers and their families, including education, can be considered as a monopsony in the labour market by the employers of labour. According to Yeh et al. [3], since the early 2000s the degree of monopsony has been sharply increasing in labour markets for manufacturing plants in the U.S. Monopsony is a constant in developing countries where labour is involved in the production of resources.

Additionally, education systems that foster a quality labour force, social security that supports workers' safety and security, and a good operating environment are basically public services provided by the government. As discussed later in the study, corporate

tax rates have been declining in all developed countries since the 1990s, and this can be considered as deemed monopsony for corporate taxes from governments.

Considering the above, it is logically consistent to tax monopsony rents and underpayments for public services by corporations and use them as a source of funding for the SDGs.

### 1.1.2. Weak Political Power of Parties Concerned and Supporters for SDG Issues

In addition to the above reason of resource allocation, there is another reason why monopsony rents from global corporations should be used to finance the SDGs: to correct the imbalance of political power.

There are many examples of social movements in which the concerned people and supporters, such as NGOs, have received new systems from governments to solve environmental and human rights problems. However, their political power is too weak to achieve sufficient results amidst the urgent need to improve global environmental problems that could affect the very survival of the human race.

To cite one numerical example, according to Drutman [4], the ratio of lobbying expenditures by corporations to those by labour unions and diffuse interest groups in the U.S. was 22:1 in 1998, 21:1 in 2001, and 34:1 in 2012 (p. 13), indicating that the political influence of SDG parties is far weaker than corporations, and the gap is growing.

Political power is weak, especially in developing countries where activism to solve problems can even put lives at risk. According to the BBC [5], 227 people (activists working to protect the environment and land rights) were killed worldwide in 2020, the highest number recorded for a second consecutive year; the report is from Global Witness.

In addition to domestic policies, the SDGs are also closely related to trade. The WTO have been concerned with trade and the environment considering the time of the reorganisation in 1995. However, the position of WTO's promoting free trade has always been at odds with its concern to ensure that environmental and poverty disparity considerations do not become non-tariff barriers in disguise. In trade negotiations, the power of lobbying activities deployed by related industries to eliminate tariffs and non-tariff barriers is overwhelming, for example, the forest produced industries during the Uruguay Round [6] and agribusiness during the Doha Round [7].

Therefore, when considering the financial resources for the SDGs, it is necessary to consider not only finding the sources of financial resources themselves, but also increasing the feasibility of the policies by letting global firms curb excessive lobbying and strengthening the political power of the SDG parties in relative terms.

### 1.2. Normative Corporate Taxation and Fiscal Science with Corporate Rent

Now we will consider the position that the surcharge of monopsony rent on the corporate income tax should serve as a financial resource for social common capital on the basis for corporate taxation.

### 1.2.1. Classical Benefit-Based Taxation and Social Costs Allocation

A description in *An Inquiry into the Nature and Causes of the Wealth of the Nation* by Smith [8] is regarded as a classical basis for the implementation of corporate income tax. Smith stated that "The subjects of every state ought to contribute towards the support of the government as nearly as possible, in proportion to their respective abilities; that is, in proportion to the revenue which they respectively enjoy under the protection of the state." (p. 777) Weinzierl [9] described it as benefit-as-ability-based taxation and stated that the classical view of benefit-based taxation was highly influential in the late 18th and early 19th centuries. Additionally, Goode [10] introduced the allocation of social costs as the basis for corporate taxation, based on the argument of Berle and Means [11].

The subject of Berle and Means' study is the distribution of profits between shareholders and managers of a corporation that has changed in nature (compared to the company that Adam Smith had in mind) and has begun to exert tremendous economic and political

power. Corporations enjoy privileges such as no inheritance and limited employee liability, and the shareholders are not liable for the debt of the corporation. "Neither the claims of ownerships nor those of control (author's notes: shareholders nor managers) can stand against the paramount interests of the community" [11] (p. 312).

Goode [10] quoted their consideration and suggested that corporations should assign their profit to the social costs of their activities as corporate income tax (p. 23). In modern terms, the financial sources of social common capital should be sourced from corporate profit.

### 1.2.2. Corporate Taxation Based on Comprehensive Income

According to Kaneko [12], in the tax theory mentioned in German finance theories from the end of the 19th century, the duty of the people to bear the tax has been highlighted, which led to the idea that the tax burden should be distributed among the people according to the tax-bearing capacity of each person (pp. 24–26). This is based on Schanz [13], who played the leading role in introducing the concept of a comprehensive income. Subsequently, it was taken over by Haig [14] and Simons [15] and went on to become the core of tax theory. Along this line, developed countries have developed income tax-centered tax systems. The concept of comprehensive income taxation has been based on maintaining vertical and horizontal equity by varying tax rates according to tax-bearing capacity. Even as European countries and Japan introduced consumption taxes in and after the 1970s, the U.S. maintained an income tax-centered tax system until the 21st century.

However, with the liberalization of capital movement and changes in corporate organisational structures, fund-raising, and investment activities, it has become necessary to consider the tax base for corporate income taxation in terms of neutrality toward corporate fund-raising and investment choices, and not promoting profit transfers to other countries. In recent years, advanced countries have been considering systems such as allowance for corporate equity (ACE), allowance for growth and investments, (AGI), and destination-based cash-flow tax (DBCFT). According to Mooij and Deuereux [16], today's corporate tax systems in developed countries allow a deduction for interest but not for equity. This leads to excessive leverage, discrimination against risky or volatile businesses, and arbitrage opportunities that erode corporate tax bases. Under this system, a portion of the normal return from equity financing is calculated from the notional interest rate and deducted from the tax base.

This system neutralises the distortionary effect of corporate taxes on the financial structure of companies. In these times of institutional transition, it would not be altogether fictitious that the tax treatment of adding a deemed monopsony rent to corporate income tax to improve distortions in resource allocation and to avoid the significant sustainability risks to which humanity is exposed.

### 1.3. International Necessity of Taxation for Rents of Corporations

Next, we will discuss the rent seeking of global corporations and their control of political power from the perspective of public choice.

### 1.3.1. Rent Seeking of Corporations

In the field of industrial organisation theory, deadweight losses caused by corporate activities that do not meet the assumption of perfect competition have long been regarded as the social cost of monopolies. In the 1950s, Harberger [17] and Schwartzmann [18] made quantitative estimates of deadweight loss. Tullock [19], one of the founders of public choice, argued that not only deadweight loss but also the excess profits earned through barriers to entry and tariffs attributed to companies are social costs. Posner [20] reviewed numerous government-regulated rent measurements.

Tullock [19] also mentioned that resources would be invested in lobbying to earn rents. Bhagwati [21] called these expenditures direct unproductive profit-seeking (DUP) activities. Monopoly and monopsony rents and transfer rents should belong to other economic agents

(consumers, governments, etc.), but when they are in the hands of the company, the funds are spent on further rent seeking.

Subsequently, this trend has continued to grow significantly in the U.S. Drutman [4] pointed out that the average lobbying expenditure of 1066 U.S. companies listed on the S&P 500 between 1981 and 2004 more than doubled in that period. More recently, U.S. corporate spending on lobbying activities increased from USD 1.13 billion in 1998 to USD 2.09 billion in 2010. This extensive lobbying is not just confined to the U.S.; corporations attempt to shape government policy in their favour in most countries (Gorostidi-Martinez et al. [22]).

Some scholars argue that corporate lobbying generates high returns. An estimation study by Alexander et al. [23] showed that lobbying activities have an astounding 22,000 percent rate of return. Another example by Etzioni [24] noted a return of approximately 6700 percent.

Therefore, especially in this economic society where companies are expanding and becoming globalized, we must be concerned that monopoly, monopsony, and transfer rents will be poured into the political activities of companies rather than into economic activities in competitive markets. They should be redistributed to appropriate attributions to avoid resource misallocations and their proliferation. Otherwise, as noted in Section 1.1.2, corporations will gain more and more of their political power compared to other types of organisations.

### 1.3.2. Global Race to the Bottom of Corporate Income Taxes

However, it is difficult to prohibit these activities because corporate lobbying tends to provide benefits to politicians with legislative authority. Even if governments could charge corporations for the unproductive rent, this might encourage companies to move overseas to avoid being taxed. Reducing tax rates has also not encouraged companies to keep their profits in their home countries. Clausing [25] and Cobham and Janský [26] provide evidence that profit shifting has grown significantly even as effective tax rates have fallen. This situation has led to the average corporate tax rate in Organization for Economic Cooperation and Development (OECD) member countries dropping considerably since the 1990s in a virtual "race to the bottom", although Ghinamo et al. [27] noted the well-known difficulty in identifying a common trend in corporate taxation from actual strategic interaction.

In recent years, increasing rates of corporate tax evasion have eroded government finances across the world. For instance, Alstadsæter et al. [28] showed that about 10 percent of global gross domestic product (GDP) is held in tax havens. Crivelli et al. [29] and Cobham and Janský [26] estimated the resulting global tax loss at between USD 650 billion and USD 5 trillion per year, respectively.

International organisations have only recently begun acting on this front. For instance, Zhu [30] stated that the "G20 + OECD" regime has taken the initiative to fight tax evasion since 2012. Although the agreement on an effective corporate tax rate of at least 15% has put an end to the "race to the bottom", it may not be enough for each country's finances to regain sound public function.

Weinzierl [31] argues that it would be easier to implement substantial international tax reforms if policymakers could point to an additional, complementary normative logic. They suggest that this is possible if the international community collaborates and calculates a normative corporate tax rate for each country—-especially for global corporations—-and works together to enforce this rate. It could be one direction in which this could be conducted to calculate a normative corporate tax rate that considers monopsony rents for multinational corporations, and to use the tax surcharge to finance social common capital.

### 1.4. Normative Corporative Income Tax with Rent and Original Attributions for Rents

This study proposes a versatile method to calculate a normative corporate tax rate that combines the monopsony rent and the corporate tax. However, it is necessary to specify the original attributing entity of each rent within the total exclusive rents.

Monopoly rents in product markets should be redistributed to consumers. This includes Schumpeter rents, which should be distributed to firms as their development and advance profits. We believe that competition policies, such as antitrust laws, have traditionally aimed for the appropriate directions. This is outside the scope of the rents that should be added to the corporate income tax this time.

The monopsony rents on input factors of production imply underpayment in labour and resource markets. However, for resource imports, the simple solutions of implementing regulatory legal systems in developed countries make little sense. In many cases, resources are supplied across national borders, and traceability is essential to prove that the extraction process is not environmentally destructive and that human rights considerations for local people are ensured. However, to guarantee sufficient traceability, it is necessary to establish a system with human resources and costs.

For example, in the system of Forest, Law, Enforcement, Governance and Trade (Flegt), the EU has agreed on Voluntary, Partnership and Agreements (VPAs) with tropical timber exporting countries to provide advice and frameworks for the establishment of such systems, and the exporting countries have set up systems to ensure traceability in the production process of tropical timber [32]. Without the groundwork for such systems, the developed countries that import timber will not be able to create an effective legal system to prohibit illegal timber imports. While building such a system, the EU implemented the EU Timber Regulation in 2013. However, only in a few cases have such systems been established.

There are many cases where workers' rights are not sufficiently guaranteed, including in developed countries, with regard to the stability of their employment opportunities and the improvement of their working environment. Equal access to quality education is also insufficient. Therefore, the underpayment of corporations should be corrected, and monopsony rents should be allocated for these capacity building and infrastructure improvements.

As noted above, the current corporate tax itself is subject to global competition due to the capital flight of global corporations. The current corporate tax "race to the bottom" is, in composition, monopsony by global corporations on multiple governments. Governments need financial resources to provide all public services, including contributions to the SDGs, although many developed country governments have budget deficits. As stated by Goode [10], monopsony rents in the "international market for corporate taxes" should be used to finance governments' provision of public services, including contributions to the SDGs.

## 2. Normative Corporate Income Tax with Rent and Its Calculation

Based on the aforementioned considerations, the corporate tax rate that should be levied is the sum of the current corporate tax rate, the monopsony rents generated in factor markets, and the type of monopsony rent yielded by global companies to the government in each country. In this study, the total of these factors is defined as a normative corporate tax rate with rent. We plan to measure the normative corporate tax rates of U.S. global companies over the last 30 years.

### 2.1. Features of the Calculation Method in This Study

Shimamoto [33] explains how to calculate the dynamic monopoly and monopsony rents for each corporation using their accounting data of Japanese companies. The objective of it was to develop a standard method for measuring rents of individual corporations from financial statement data and to analyse the relationship between rents and political spending and R&D expenditures. The variables were the same as this U.S. study for operating expenses (w1v1), non-operating expenses and extraordinary losses (w2v2), and corporate income taxes (w3v3). We did not assume deemed monopsonies for corporate income taxes, and we set the markup rate of zero for w3v3 at [33]. In this issue, we focus on the monopsonies for resources and labour by global firms and the underpayment for public services resulting from their lower corporate taxes, which have led to the emergence of

rents. The firms are using these goods and services as production inputs, and their optimal prices should be at least the marginal factor costs. The fact that the markup ratio is positive means that a rent is being generated. The corporate tax markup rate is not set at zero. We calculate the time series of monopsony rents to calculate the normative corporate tax rate in this study. The flow from previous research and the features of this model are as follows.

### 2.1.1. Three Streams of Previous Research

The first stream is the measurement of rent in public choice. Many empirical studies regarding rent owing to government regulations have been conducted since rent seeking was pointed out by Tullock [19]. However, in more recent studies, for example, Tarr [34], Salhofer et al. [35], and Jarvis [36], these have been estimated generally by measuring producer surplus at the industrial level and through aggregated variables. This is only a rough basis for analysing taxation systems; thus, actual measures such as taxation and surrogacy for individual companies are needed.

The second stream is the measurement of the degree of monopoly (also called markup rates) and monopsony for factor markets. Econometric methods of simultaneous equation have been used to estimate them, as in Appelbaum [37], Schroeter [38], and Azzam and Pagoulatos [39]. However, when trying to obtain them from the time series data of an individual company, we often face multicollinearity issues.

The third stream is total factor productivity (TFP). Since Solow [40] proposed a neoclassical growth model that multiplies a linear homogeneous function with capital and labour as production factors by a technology coefficient, many estimates of TFP have been made using this functional form. In recent years, it has also been widely used in the economic analysis of various countries. A Cobb–Douglas production function is used to estimate TFP. Eventually, the Solow residual, which is the technical coefficient, was thought to include not only technological progress but also the markup rate due to imperfect competition, and research was conducted to separate it. Hall [41] used instrumental variables as a method to estimate it, and Roeger [42] constructed a method that can estimate the markup rate using Ordinary Least Squares from the difference between the primal and dual Solow residuals owing to imperfect competition. Furthermore, Bottini and Molnár [43] introduced a scale factor exogenously and used Roeger's method to estimate the markup rates of 20 OECD countries using accounting data. In recent years, Roeger's method has been commonly used. For example, the monthly report of the Deutsche Bundesbank [44], which reported markup rates for seven EU countries.

### 2.1.2. Differences from TFP Method

The approach of this study is similar to the third method. However, the purpose of this research is not to estimate the markup rate itself but to calculate the rent amount by multiplying it by the input output level. For that reason, it is necessary to align the markup rate and the production point for each term as a set. Thus, this study used the method of obtaining these estimates by solving simultaneous equations using the mathematical programming method as presented by Shimamoto [33].

In addition to the assumptions commonly used in TFP, this study assumes that long-term interest rates are used as the discount rate, which also means the marginal efficiency of capital, to calculate the rent amount. This study analyses the taxation of each firm; the production function should comprehensively cover production activities, so the factor of production is set to operating expenses, non-operating expenses, extraordinary losses, and corporate tax.

### 2.1.3. Pseudo-Competitive Long-Term Equilibrium Model

The basic idea of the modelling in this study is as follows. It is assumed that the target producer is a monopolist in the product market and a monopsonist in the production factor markets. In other words, the production volume and price, factor input volumes, and factor prices that a company experienced in each past period should be at the monopolistic

equilibrium point. However, now it is assumed that this past production volume and factor input volume are established as a competitive equilibrium under the same production technology of this company. At this time, the competitive equilibrium price of the product should be equal to the marginal revenue and marginal cost in the production volume of the product. The difference between these variables and the monopoly price is the markup ratio of the product market. Similarly, the competitive equilibrium price of the factor of production under the same production technology of this company is equal to the marginal factor costs and marginal revenue products in this factor input volume. The differences between these variables and the monopsonic prices are the markup ratios of the production factor markets. Ultimately, the objective function is to maximise the total discounted present value of the company's profit time series. We calculate a time series of rents (the degree of monopoly and monopsony) in which the capital quantity time series of this company realised in the past maximises this objective function.

*2.2. Model*

The specific model is as follows.

2.2.1. Short-Term Equilibrium Conditions

We assume monopolistic and monopsonistic product markets. Producers maximise short-term profits and achieve long-term equilibrium. We assume that a producer uses a general Cobb–Douglas production technology to produce one product using four production factors. This function is denoted by

$$y = \alpha_5 v_1^{\alpha 1} v_2^{\alpha 2} v_3^{\alpha 3} K^{\alpha 4}, \tag{1}$$

where $y$ is the output quantity, $v_1, v_2$, and $v_3$ are the quantities of variable inputs, and $K$ is the quantity of the fixed input, namely capital. In the short-term equilibrium, $K$ is a given value, and $\alpha_1 + \alpha_2 + \alpha_3 < 1$, which ensures that the marginal cost function is convex.

As the producer simultaneously faces both a monopolistic product market and monopsonistic factor market, the short-run profit maximisation problem is given as

$$\text{Max } \pi_m = p(y) \cdot y - w_1(v_1) \cdot v_1 - w_2(v_2) \cdot v_2 - w_3(v_3) \cdot v_3 - rK \tag{2}$$
$$\text{s.t. (1).}$$

The optimal conditions come from differentiating the Lagrange equation by the variable factors $v_1, v_2$, and $v_3$. These are given as follows:

$$\{p'(y) \cdot y + p(y)\} \cdot f'_{v_i} = \{w'_i(v_i) \cdot v_i + w_i(v_i)\}, \ i = 1, \ 2, \ 3. \tag{3}$$

$\{p'(y) \cdot y + p(y)\}$ can be expressed as $(1 + \gamma) \cdot p(y)$, where $\gamma$ is now assumed to have a constant inverse demand elasticity and $-1 < \gamma \leq 0$. $p_m$(y) can be defined by

$$p_m(y) = \{p'(y) \cdot y + p(y)\} = (1 + \gamma) \, p(y). \tag{4}$$

$\{w'_i(v_i) \cdot v_i + w_i(v_i)\}$ can be expressed as $(1 + \sigma_i) \cdot w_i$, where $\sigma_i$ is now assumed to have a constant inverse factor supply elasticity and $\sigma_i \geq 0$. In the same way, $w_{im}(v_i)$ can be defined by

$$w_{im}(v_i) = \{w'_i(v_i) \cdot v_i + w_i(v_i)\} = (1 + \sigma_i) \cdot w_i(v_i), \ i = 1, \ 2, \ 3. \tag{5}$$

The short-run optimisation conditions in the monopoly and monopsony markets can be expressed by arranging Equations (1), (3) and (4) as follows:

$$y = \alpha_5 \cdot \left( \frac{w_{1m}^{1-\alpha_2-\alpha_3} \cdot w_{2m}^{\alpha_2} \cdot w_{3m}^{\alpha_3}}{\alpha_5 \cdot \alpha_1^{1-\alpha_2-\alpha_3} \cdot \alpha_2^{\alpha_2} \cdot \alpha_3^{\alpha_3} \cdot p_m K^{\alpha_4}} \right)^{\frac{\alpha_1}{\alpha_1 + \alpha_2 + \alpha_3 - 1}} \cdot$$

$$\left( \frac{w_{1m}^{\alpha_1} \cdot w_{2m}^{1-\alpha_1-\alpha_3} \cdot w_{3m}^{\alpha_3}}{\alpha_5 \cdot \alpha_1^{\alpha_1} \cdot \alpha_2^{1-\alpha_1-\alpha_3} \cdot \alpha_3^{\alpha_3} \cdot p_m K^{\alpha_4}} \right)^{\frac{\alpha_2}{\alpha_1 + \alpha_2 + \alpha_3 - 1}} \cdot \left( \frac{w_{1m}^{\alpha_1} \cdot w_{2m}^{\alpha_2} \cdot w_{3m}^{1-\alpha_1-\alpha_2}}{\alpha_5 \cdot \alpha_1^{\alpha_1} \cdot \alpha_2^{\alpha_2} \cdot \alpha_3^{1-\alpha_1-\alpha_2} \cdot p_m K^{\alpha_4}} \right)^{\frac{\alpha_3}{\alpha_1 + \alpha_2 + \alpha_3 - 1}} \cdot \quad (6)$$

$$K^{\alpha_4}$$

$$v_i = \alpha_5 \cdot \left( \frac{w_{im}^{1-\alpha_j-\alpha_k} \cdot w_{jm}^{\alpha_j} \cdot w_{km}^{\alpha_k}}{\alpha_5 \cdot \alpha_i^{1-\alpha_j-\alpha_k} \cdot \alpha_j^{\alpha_j} \cdot \alpha_k^{\alpha_k} \cdot p_m K^{\alpha_4}} \right)^{\frac{1}{\alpha_1 + \alpha_2 + \alpha_3 - 1}} , \, i, \, j, \, k = 1, 2, 3 \, (i \neq j \neq k) \quad (7)$$

It is important to note that these equations are not normal supply and factor demand functions: $p_m(y)$ and $w_{im}(v_i)$ are endogenous variables and differ from the exogenous prices of a product and factors in competitive markets. This interdependence makes it difficult to find an optimal point and formulate an empirical model.

We utilised the relationship between imperfect competition models and perfect competition models to facilitate our calculations. Now $y^{t*}$ indicates the short-term optimum production level and $p^{t*}$ is the equilibrium price in this imperfect competition model as described in Figure 1. A superscript $t$ indicates a value in the $t$ period.

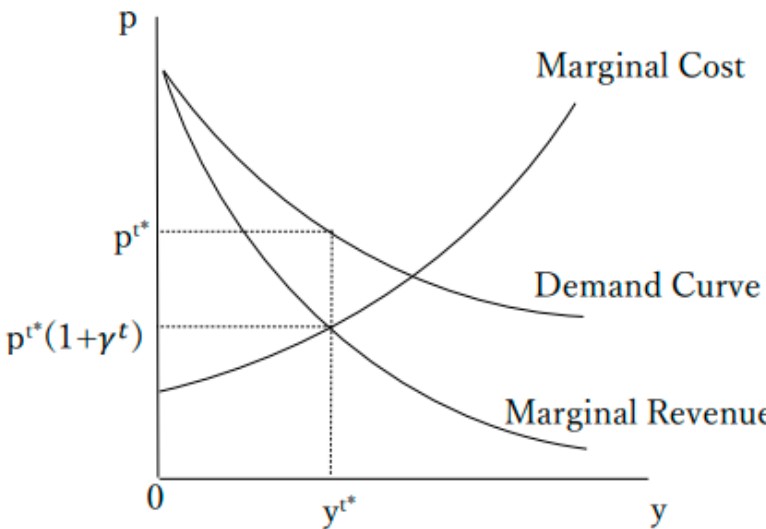

**Figure 1.** Imperfect competition and perfect competition in the products market.

Given the same production technology and the same given value of $K^t$, $y^{t*}$ is also the short-term equilibrium production volume under the given market price $p^{t*}(1 + \gamma^t)$ in perfect competition. $\gamma^t$ is the degree of monopoly in this imperfect competition model at the optimal point.

We can consider the production factor markets in the same manner. In this model, the producer is a monopsonist in the factor markets. $v_1^{t*}, v_2^{t*}$, and $v_3^{t*}$ are the short-term optimal factor quantities, and $w_1^{t*}, w_2^{t*}$, and $w_3^{t*}$ are the equilibrium prices in this monopsonic equilibrium, as described in Figure 2. Under the same production technology and the same given value of $K^t$, $v_1^{t*}, v_2^{t*}$, and $v_3^{t*}$ are also the short-term optimal factor quantities under the given market prices $w_1^{t*}(1 + \sigma_1^t), w_2^{t*}(1 + \sigma_2^t)$, and $w_3^{t*}(1 + \sigma_3^t)$ in perfect competition. Then, we can get markup rates of $\sigma_1^t, \sigma_2^t$, and $\sigma_3^t$, which satisfy the condition where $w_1^{t*} v_1^{t*}, w_2^{t*} v_2^{t*}$, and $w_3^{t*} v_3^{t*}$ are the costs for each realised by profit maximisation in imperfect competition in the $t$ period.

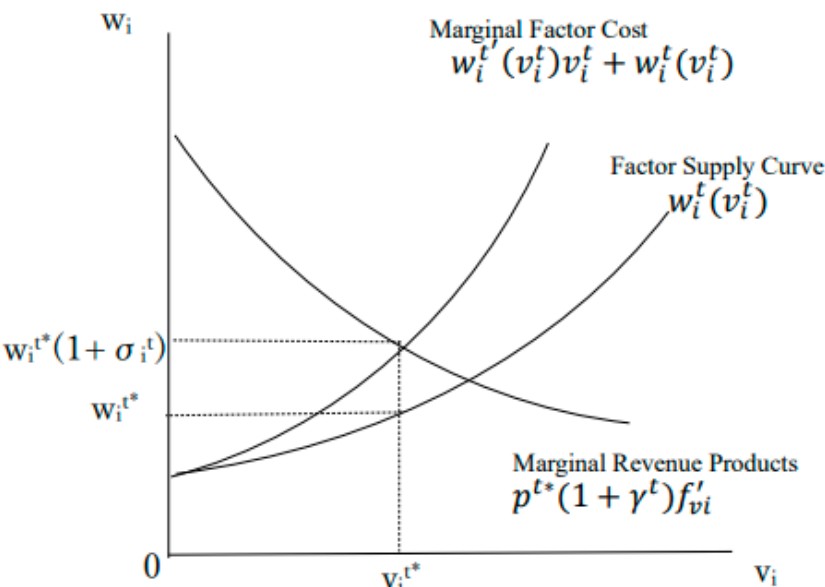

**Figure 2.** Imperfect competition and perfect competition in the factor market.

Therefore, considering the long-term equilibrium conditions, the supply Equation (6) and factor demand Equation (7) can be regarded as the supply function and factor demand function, respectively, under the given $K^t$, $p^{t*}(1 + \gamma^t)$, and $w_i^{t*}(1 + \sigma_i^t)$, which satisfy the short-term equilibrium conditions in the perfectly competitive model.

2.2.2. Long-Term Equilibrium Conditions

Long-term equilibrium conditions are derived by maximising the time series total of the discounted present value of profits defined by the short-term pseudo-competitive equilibrium model minus capital costs. By deriving these long-term pseudo-competitive profit maximisation conditions, we can find the optimal $\gamma^t$, $\sigma_1^t$, $\sigma_2^t$, and $\sigma_3^t$ that makes the time series data of $K^t$, $p^{t*}y^{t*}$, $w_1^{t*}v_1^{t*}$, $w_2^{t*}v_2^{t*}$, and $w_3^{t*}v_3^{t*}$ for each of the four periods the optimal dynamic solution. These past data can be regarded as results that have satisfied both the short-term and long-term equilibrium conditions. We can then determine the rent ratios, $\gamma^t$ and $\sigma_i^t$, which are the proportions of rent in the product price and factor prices, respectively.

Now, let us formulate the long-term pseudo-competitive profit maximisation conditions. The conditions are maximised for discrete time periods from 1 to $T$. The pseudo-competitive profit function in period $t$ is defined as follows:

$$
\begin{aligned}
\pi^t = {} & p^{t*}(1 + \gamma^t) \cdot y^t \left( p^{t*}(1 + \gamma^t), w_1^{t*}(1 + \sigma_1^t), w_2^{t*}(1 + \sigma_2^t), w_3^{t*}(1 + \sigma_3^t), K^t \right) \\
& - w_1^{t*}(1 + \sigma_1^t) \cdot v_1^t \left( p^{t*}(1 + \gamma^t), w_1^{t*}(1 + \sigma_1^t), w_2^{t*}(1 + \sigma_2^t), w_3^{t*}(1 + \sigma_3^t), K^t \right) \\
& - w_2^{t*}(1 + \sigma_2^t) \cdot v_2^t \left( p^{t*}(1 + \gamma^t), w_1^{t*}(1 + \sigma_1^t), w_2^{t*}(1 + \sigma_2^t), w_3^{t*}(1 + \sigma_3^t), K^t \right) \\
& - w_3^{t*}(1 + \sigma_3^t) \cdot v_3^t \left( p^{t*}(1 + \gamma^t), w_1^{t*}(1 + \sigma_1^t), w_2^{t*}(1 + \sigma_2^t), w_3^{t*}(1 + \sigma_3^t), K^t \right) \\
& - Q^t \cdot I^t \left( K^{t-1}, K^t, \delta^t \right),
\end{aligned}
\tag{8}
$$

where $y^t(\cdot)$ *and* $v_i^t(\cdot)$ are defined by Equations (6) and (7), and $\gamma^t$, $\sigma_1^t$, $\sigma_2^t$, and $\sigma_3^t$ are assumed to change over time. Investment in period $t$, $I^t\left(K^{t-1}, K^t, \delta^t\right)$ is defined as follows:

$$
I^t\left( K^{t-1}, K^t, \delta^t \right) = K^t - \left( 1 - \delta^t \right) \cdot K^{t-1},
\tag{9}
$$

where $\delta^t$ is the depreciation rate in period $t$, and $Q^t$ is the exogenous unit price of investment in period $t$.

The long-term equilibrium condition arises from maximising the sum of the discounted present value of $\pi^t$ from period 1 to $T$ based on $K^t$, as follows:

$$\max_{K^t}\Pi = \pi^1 + \sum_{t=2}^{T}\prod_{s=2}^{t}\frac{1}{(1+r^s)}\pi^t. \tag{10}$$

Thus, the necessary condition for optimisation, is given as follows:

$$\begin{aligned}\frac{\partial\Pi}{\partial K^t} = \prod_{s=2}^{t}\frac{1}{(1+r^s)}\cdot\Big[p^{t*}\big(1+\gamma^t\big)\cdot\frac{\partial y^t}{\partial K^t} - w_1^{t*}\big(1+\sigma_1^t\big)\cdot\frac{\partial v_1^t}{\partial K^t} - w_2^{t*}\big(1+\sigma_2^t\big)\cdot\frac{\partial v_2^t}{\partial K^t} \\ -w_3^{t*}\big(1+\sigma_3^t\big)\cdot\frac{\partial v_3^t}{\partial K^t} - Q^t\cdot\frac{\partial I^t}{\partial K^t}\Big]+\prod_{s=2}^{t+1}\frac{1}{(1+r^s)}\Big[-Q^{t+1}\cdot\frac{\partial I^{t+1}}{\partial K^t}\Big] = 0.\end{aligned} \tag{11}$$

Using $\frac{\partial B}{\partial A} = \frac{\partial B}{\partial \ln B}\cdot\frac{\partial \ln B}{\partial \ln A}\cdot\frac{\partial \ln A}{\partial A}$ and $\partial \ln C/\partial C = 1/C$ that generally hold for logarithmic differentiation, the following equations are derived from Equations (6) and (7) which are the short-term optimising conditions.

$$\frac{\partial y}{\partial K} = \frac{-\alpha_4}{\alpha_1+\alpha_2+\alpha_3-1}\cdot\frac{y}{K} \tag{12}$$

$$\frac{\partial v_i}{\partial K} = \frac{-\alpha_4}{\alpha_1+\alpha_2+\alpha_3-1}\cdot\frac{v_i}{K}, \qquad i = 1, 2, 3 \tag{13}$$

Using Equations (12) and (13), Equation (11) can be finally arranged into the following simple equation:

$$\begin{aligned}\big(1+\gamma^t\big)\cdot\frac{p^t y^t}{K^t} - \big(1+\sigma_1^t\big)\cdot\frac{w_1^t v_1^t}{K^t} - \big(1+\sigma_2^t\big)\cdot\frac{w_2^t v_2^t}{K^t} - \big(1+\sigma_3^t\big)\cdot\frac{w_3^t v_3^t}{K^t} \\ -\frac{\alpha_1+\alpha_2+\alpha_3-1}{-\alpha_4}\cdot\Big[Q^t - \frac{1}{(1+r^{t+1})}\cdot Q^{t+1}\cdot\big(1-\delta^{t+1}\big)\Big] = 0.\end{aligned} \tag{14}$$

(See Shimamoto [33] for the detailed derivation process. In this empirical study, the data are collected from the financial statements of each company. The depreciation amount for a year is included as an item among the costs. Therefore, we set $\delta^{t+1}$ as zero to avoid double counting.)

### 2.3. Ingenuity to Apply Corporate Accounting Data to Production Functions for Calculation

Here, we calculate the normative corporate income tax rate in a time series based on which we calculated the rent by applying the model to corporations' financial data.

The output value ($py$) is the total sales, and the four production factors $w_1^t v_1^t, w_2^t v_2^t, w_3^t v_3^t$, and $K$ are operating expenses, non-operating expenses plus extraordinary losses, corporate taxes, and total assets. Among the model's four parameters—$\gamma^t, \sigma_1^t, \sigma_2^t$, and $\sigma_3^t$—one should be removed when $R$ (= $\frac{\alpha_1+\alpha_2+\alpha_3-1}{-\alpha_4}$) is regarded as a fourth variable. At this time, $\sigma_2$ will be set to 0 because of the nature of the production factor without the necessity of monopsony.

When we regard the variable $R$ relating to scale as the fourth parameter, $\gamma, \sigma_1, \sigma_3$, and $R$ can be obtained by quadratic programming using data for at least four years, as calculated in Shimamoto [33]. However, when the parameters are determined in this way, the time series values of $R$ fluctuate drastically from year to year, which makes the sequential values of rent unstable. Therefore, we define a scale variable, $S$, as follows:

$$S = \alpha_1 + \alpha_2 + \alpha_3 + \alpha_4. \tag{15}$$

The calculation is performed by setting $S$ to 1 (i.e., $R$ = 1), and sensitivity analysis will be performed with $S$ = 1.2, which can be regarded as the upper bound of induced returns to scale, as the return to scale in the U.S. in recent years was calculated to be 1.12 according to Boussemart et al. [45].

In the case of $S \neq 1$, when trying to specify the value of $R$, it is necessary to give the value of $\alpha_4$ as well. This represents the effect of increasing $K$ on production when other variable factors are constant. Originally, empirical research was required to identify this

value, but in general, a fairly large-scale study is required to identify this value. This time, the sensitivity analysis is performed when the value of $\alpha_4$ is 0.5 ($R = 0.6$) and 0.9 ($R = 0.78$).

*2.4. Capital Reward and Rent Distribution*

In the case of constant returns to scale (i.e., $S = 1$), total rent is regarded as $-\gamma^t p^t y^t + \sigma_1^t w_1^t v_1^t + \sigma_2^t w_2^t v_2^t + \sigma_3^t w_3^t v_3^t$. However, if the yield shows increasing returns to scale, it is necessary to devise profit and rent distribution. To explain this point, we must define the distribution of capital reward and rent specifically.

Transforming Equation (14),

$$
\begin{aligned}
&\frac{p^t y^t}{K^t} - \frac{w_1^t v_1^t}{K^t} - \frac{w_2^t v_2^t}{K^t} - \frac{w_3^t v_3^t}{K^t} \\
&= R \cdot \left[ Q^t - \frac{1}{(1+r^{t+1})} \cdot Q^{t+1} \cdot (1 - \delta^{t+1}) \right] + \frac{-\gamma^t p^t y^t}{K^t} + \frac{\sigma_1^t w_1^t v_1^t}{K^t} + \frac{\sigma_3^t w_3^t v_3^t}{K^t}.
\end{aligned}
\tag{16}
$$

In the accounting data, capital depletion $\delta$ is recorded as an item in $w_1^t v_1^t$ as a depreciation expense, so $\delta = 0$. Further, regarding $I^t$ as the investment amount, both $Q^t$ and $Q^{t+1}$ are equal to 1. Taking these things into account, we multiply both sides of Equation (16) by $K^t$:

$$
\begin{aligned}
&p^t y^t - w_1^t v_1^t - w_2^t v_2^t - w_3^t v_3^t \\
&= R \cdot \frac{r^{t+1}}{1+r^{t+1}} \cdot K^t + (-\gamma^t p^t y^t + \sigma_1^t w_1^t v_1^t + \sigma_2^t w_2^t v_2^t + \sigma_3^t w_3^t v_3^t).
\end{aligned}
\tag{17}
$$

Since $r^{t+1}$ is the interest rate, it can be generally regarded as (discounted present value of) the marginal efficiency of capital. When $R = 1$, $R \cdot \frac{r^{t+1}}{1+r^{t+1}} \cdot K^t$ means a competitive and normative capital reward (This is equal to the equilibrium interest rate when the capital markets are competitive, which means that there is no surplus in the capital markets). Thus, this equation can be interpreted as:

$$\text{Net Income = Capital Reward + Rent.} \tag{18}$$

In other words, in the case of constant returns to scale, rewards above the marginal efficiency of capital are rents, and the part that should be added to the corporate tax rate when we obtain the normative corporate tax rate is the part of this rent excluding the monopoly rent of the product market $-\gamma^t p^t y^t$ because monopoly rents on the product market include Schumpeter rents and the rest are excessive gifts from consumers.

What about the case of increasing returns on scale? When $S > 1$, $R < 1$ and $R$ converge to 0 as S increases. If we consider $R \cdot \frac{r^{t+1}}{1+r^{t+1}} \cdot K^t$ as the capital reward according to Equation (17) in this case, as the scale harvest increases, the proportion of capital rewards among net income decreases, and the proportion of rent increases.

Therefore, when S > 1, the capital reward is set to $S \cdot \frac{r^{t+1}}{1+r^{t+1}} \cdot K^t$ to reflect economies of scale. Equation (19) is obtained by dividing Equation (17) into this capital reward term and the rest into the rent portion. In this equation, the first term on the right-hand side is capital remuneration, and the second and subsequent terms are considered to be rents.

$$
\begin{aligned}
\text{Net Income} &= S \cdot \frac{r^{t+1}}{1+r^{t+1}} \cdot K^t \\
&+ \left\{ \left( -\gamma^t p^t y^t + \sigma_1^t w_1^t v_1^t + \sigma_2^t w_2^t v_2^t + \sigma_3^t w_3^t v_3^t \right) + R \cdot \frac{r^{t+1}}{1+r^{t+1}} \cdot K^t - S \cdot \frac{r^{t+1}}{1+r^{t+1}} \cdot K^t \right\}.
\end{aligned}
\tag{19}
$$

Additionally, in this case, when calculating the normative corporate tax rate, we use the total rent minus the monopoly rent of the product market.

*2.5. Data and Calculation*

All the data used for this simulation can be found through the following link. https://data.mendeley.com/datasets/ds22hpy629/2, accessed on 4 February 2023.

The system of equations was solved using MATLAB software (These equation systems were solution for quadratic programming using the *lsqlin* in MATLAB, which solve a constrained linear least-squares problem). Using matrix expressions, simultaneous equations that solve for $\gamma$, $\sigma_1$, and $\sigma_3$ in Equation (14) can be specified and then estimated with three years of financial data as follows:

$$\begin{bmatrix} R\hat{Q}^t + \hat{V}_2^t \\ RQ^{\hat{t}+1} + V_2^{\hat{t}+1} \\ RQ^{t+2} + V_1^{\hat{t}+2} \end{bmatrix} = \begin{bmatrix} \hat{Y}^t & -\hat{V}_1^t & -\hat{V}_3^t \\ Y^{\hat{t}+1} & -V_1^{\hat{t}+1} & -V_3^{\hat{t}+1} \\ Y^{\hat{t}+2} & -\hat{V}_1^{t+2} & -V_3^{\hat{t}+2} \end{bmatrix} \begin{bmatrix} (1+\gamma^t) \\ (1+\sigma_1^t) \\ (1+\sigma_3^t) \end{bmatrix}, \tag{20}$$

where $\hat{Q}^t \equiv Q^t - \frac{1}{1+r^{t+1}} \cdot Q^{t+1}$, $\hat{Y}^t \equiv \frac{p^t y^t}{K^t}$, $\hat{V}_i^t \equiv \frac{w_i^t v_i^t}{K^t}$. $\gamma^t$, and $\sigma_i^t$ ($i = 1,3$) are the markup rates for outputs and inputs in periods $t$ to $t+2$. By these equations, we obtain $\gamma$, $\sigma_1$, and $\sigma_3$.

We used 36 years' worth of financial data and set $t$ from the first year to the 34th year. However, the equation for period $t$ contains the discount rate of the $t + 1$ period, $r^{t+1}$, so the maximum length of the time series of solutions of $\gamma$, $\sigma_1$, and $\sigma_3$ is 33.

We collected the accounting data of 234 corporations in the U.S. S&P 500 in 2018 (downloaded from Mergent online). These companies were classified into 28 industries according to their Standard Industrial Classification (SIC) codes. (While the numbering followed SIC codes wherever possible, several sectors that had few firms were consolidated. For this reason, there are numbers up to 33, but there are only 28 industries.) Data on discount rates are U.S. interest rates for each year, as found in the International Monetary Fund's *International Financial Statistics*. The unit price of investment ($Q$) was set as one for every year.

## 3. Results

*3.1. Differences between Corporate Tax Rates and Normative Corporate Tax Rates*

How would the corporate income tax rate change if the rent calculated in this study were added to the corporate income tax? Table 1 shows a comparison between the ratio of corporate income tax to profit before tax and the ratio of corporate income tax plus rent, excluding monopoly rent in product markets, to profit before tax for an average of 33 years for 28 industries. The results show that the banking and insurance industries have almost zero rents, while the rest of the industries, not only manufacturing but also services, have rents ranging from around 10% to as high as 20% of pre-tax profits on average over the 33-year period. Furthermore, the manufacturing sector does not have a particularly high rent ratio. The SDGs should be paid not only by the manufacturing sectors, but also by the service sectors. However, in this model, it is not possible to decompose resource and labour rents (Yeh et al. [3] examined monopsony for the U.S. manufacturing labour markets in a sophisticated panel data analysis, and found that the baseline value of the markdown rate for production workers in the manufacturing sector, which is called the markup rate in this study, was 1.364, while that for non-production workers was 1.53. This gives us the inference that the labour market is one of the reasons why the rent ratio is not low in the non-manufacturing sector in this study).

**Table 1.** Corporate income tax including rent.

| Industry | W3/TI | S = 1<br>(W3 + Rent)/TI | S = 1.2 α4 = 0.5<br>(W3 + Rent)/TI | S = 1.2 α4 = 0.9<br>(W3 + Rent)/TI |
|---|---|---|---|---|
| 2 Mining | 0.324 | 0.512 | 0.421 | 0.424 |
| 3 Construction | 0.362 | 0.563 | 0.479 | 0.468 |
| 4 Food, tobacco | 0.325 | 0.500 | 0.451 | 0.446 |
| 5 Textile, fabric, apparel | 0.342 | 0.532 | 0.453 | 0.457 |
| 6 Lumber, furniture, paper | 0.350 | 0.515 | 0.456 | 0.463 |
| 7 Printing, publishing | 0.134 | 0.420 | 0.340 | 0.282 |
| 8 Chemical | 0.290 | 0.496 | 0.454 | 0.452 |
| 9 Petroleum refining | 0.337 | 0.627 | 0.524 | 0.485 |
| 10 Rubber, miscellaneous plastics | 0.347 | 0.589 | 0.516 | 0.516 |
| 12 Stone, clay, glass, concrete | 0.230 | 0.308 | 0.310 | 0.309 |
| 13 Primary metal | 0.352 | 0.609 | 0.519 | 0.472 |
| 14 Fabricated metal products (except machinery, transportation equipment | 0.310 | 0.513 | 0.433 | 0.438 |
| 15 Industrial, commercial machinery, and computer equipment | 0.216 | 0.425 | 0.367 | 0.357 |
| 16 Electronic machinery except computers | 0.298 | 0.498 | 0.434 | 0.427 |
| 17 Transportation equipment | 0.310 | 0.464 | 0.416 | 0.412 |
| 18 Measuring, analyzing, controlling equipment | 0.302 | 0.509 | 0.452 | 0.443 |
| 19 Miscellaneous manufacturing industries | 0.266 | 0.457 | 0.438 | 0.440 |
| 20 Railroad, motor freight, postal service, airline, transportation | 0.361 | 0.514 | 0.441 | 0.421 |
| 21 Water transportation, gas, electric, sanitary services | 0.339 | 0.442 | 0.460 | 0.315 |
| 22 Communications | 0.323 | 0.444 | 0.414 | 0.399 |
| 23 Wholesale trade | 0.360 | 0.535 | 0.464 | 0.461 |
| 24 Retail trade | 0.386 | 0.592 | 0.552 | 0.553 |
| 25 Banking | 0.302 | 0.360 | 0.305 | 0.300 |
| 26 Insurance | 0.346 | 0.514 | 0.375 | 0.340 |
| 28 Real estate | 0.293 | 0.529 | 0.477 | 0.462 |
| 29 Health services | 0.398 | 0.620 | 0.551 | 0.543 |
| 30 Business services | 0.343 | 0.567 | 0.522 | 0.520 |
| 33 Public administration | 0.354 | 0.645 | 0.590 | 0.531 |
| Average | 0.318 | 0.511 | 0.450 | 0.433 |

Note: W3 = corporate income tax, TI = profit before tax
rent = rent excluding monopoly rent of product markets

*3.2. Time Series of Corporate Tax Rate and Normative Corporate Tax Rate*

Figure 3 shows the time series of corporate income tax plus rent, excluding monopoly rent of product markets and corporate income taxes, averaged across 234 companies. It shows the time series of the normative tax rate under the various assumptions of scale variables S and $\alpha_4$. It shows that the corporate tax rate has fallen gradually over the past 30 years while the rent rate has risen monotonically. Furthermore, the rent rate rose at roughly the same pace or slightly faster than the decline in the corporate tax rate. Based on the above, we can see that the normative corporate tax rate for the past 30 years has been stable at approximately 50 percent, with a slight upward trend since the 1990s, when S = 1.2.

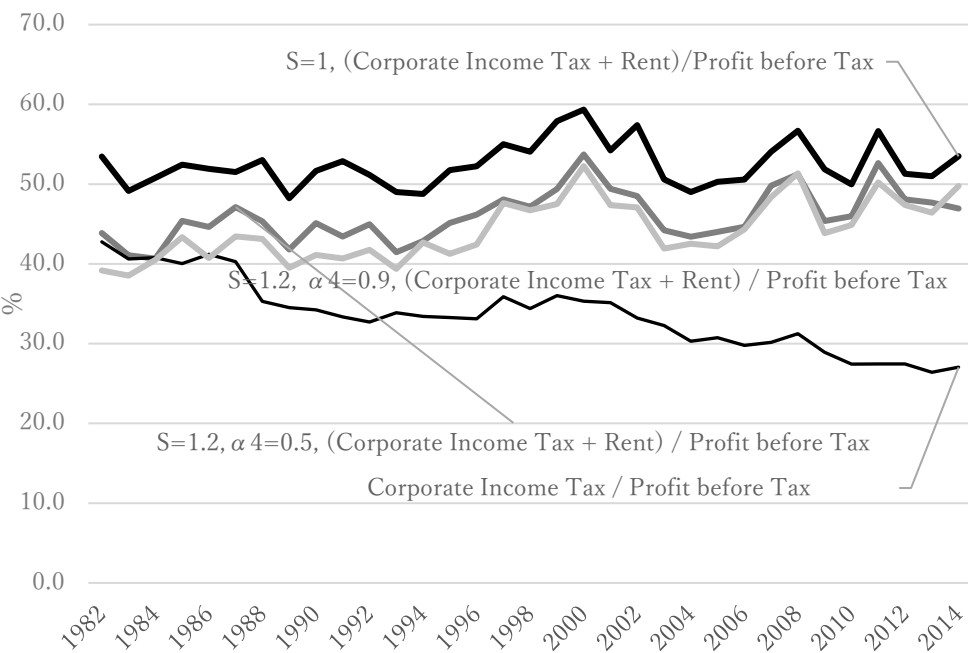

Note: rent is total rent minus exclusive rent in the product market
S is variable indicating returns to scale

**Figure 3.** Time series of rent and corporate income tax.

Based on an approximate observation of the results, we find that since the 1980s, the corporate tax rates paid by global corporations in the U.S. have declined, and that the global corporations have gained rents almost as much as the U.S. government has lost tax revenues that they should have spent on public services. Therefore, the amount lost due to the decline in the corporate tax rate can be surcharge for to finance the SDGs.

## 4. Discussion and Conclusions

### 4.1. Key Findings and Policy Recommendations

In this study, we have considered the question of how the SDGs, in economic terms, social common capital, should be financed. The double dividend of environmental taxes has limited versatility in terms of their regressivity and the application of only emission issues. The SDGs are divided into environmental issues and human issues. Both areas are issues related to inputs to the production process of global companies. They do not provide sufficient consideration to the environment and human rights. Consequently, they enjoy monopsony rents due to cheap resource prices and wages. Additionally, they have overwhelming political power compared to the people concerned and supporters that appeal for improvement of the problems. Corporate tax rates have declined in recent years and payments for public services are not considered enough. Therefore, we propose that taxation of monopsony rents and underpayments for public services of global corporations is necessary.

Then, how much are the actual normative corporate tax rates with rents for global corporations? Calculations were attempted for the U.S. case. The average normative corporate income tax rate for the 234 U.S. S&P 500 companies for 33 years starting from 1982 was calculated using mathematical programming with assumptions close to TFP. The key is the new method for calculating the degrees of monopoly and monopsony (also called markup ratios) of a company in a pseudo-competition model.

The average corporate income tax rate for these companies gradually dropped from 42.8% in 1982 to 27% in 2014 according to the data. However, the calculated rents have almost compensated for or slightly exceeded the decrease in the corporate income tax

rate, depending on the scenario. Obviously, the normative corporate income tax rate has remained flat at approximately 40% to 60%, depending on values of parameters. The U.S. government lost the opportunity to spend that amount on the social common capital, because it was distributed to the global corporations as their rents.

This is a common problem in developed countries. By conducting similar empirical studies in other major developed countries, it will be possible to set corporate tax rate targets that should be coordinated internationally.

*4.2. Limitations and Future Prospects*

There are four limitations to this study. First, the discount rate (= marginal efficiency of capital) is set to the U.S. long-term interest rate at this time. However, considering the value of the marginal efficiency of capital directly affects the normative corporate tax rate, a sensitivity analysis should be conducted. Second, the scale factor was based on an estimated value in the U.S. by Boussemart et al. [45]. Since the scale factor for each company or each industry can be calculated by Boussemart's method using accounting data, the value should be used for rent calculation. Third, our model relies on the assumption that the price elasticity of demand faced by each firm is constant at least in the neighbourhood of each period's data. While this assumption is essential to the calculation results at this time, we would like to explore ways to loosen this assumption in the future. Fourth, in the present model, the value of $\alpha_4$ cannot be determined endogenously when $S \neq 1$. For the case of $S = 1.2$, rents were calculated for $\alpha_4 = 0.5$ and 0.9, and sensitivity analysis was performed. As far as Figure 3 shows, the results were not significantly affected when we compared to the value of the scale variable S. However, it has not been clarified whether this was due to the structure of the model or the result was dependent on the characteristics of the sample data. We would like to further investigate this point. It is desirable to identify the values of parameters by either obtaining them exogenously or endogenously, if possible.

A more fundamental issue is whether tax revenues can be distributed correctly. We have argued that the government should tax monopsony rent of global corporations to funding for social common capitals. However, even if the developed countries cooperate to raise their corporate income tax rates, ensuring the proper distribution of the tax revenue among these public services will be another critical issue.

In the current form of democracy, it is known in the theory of social choice that under the current election system, politicians who prioritise the interests of enthusiastic support groups and voters are likely to be elected. It is thought that such a political environment has accelerated the lobbying of global companies. As a result, in recent years, the corporate tax rates of each country have suffered from a "race to the bottom." Given the above, it can be said that it is necessary to improve the political system, especially the electoral system, even in developed countries to realise sustainable and highly equal resource allocation. We hope that further research on election rules such as Okamoto and Sakai [46] will be conducted.

Based on this study, it is necessary to accumulate similar analyses for other countries, which could lead to international cooperation among countries in setting guidelines for an appropriate corporate income tax rate, especially for global companies.

**Funding:** This research was funded by JSPS KAKENHI, grant number: 16K00685.

**Institutional Review Board Statement:** Not applicable.

**Informed Consent Statement:** Not applicable.

**Data Availability Statement:** The data presented in this study are openly available in Mendeley Data at https://data.mendeley.com/datasets/ds22hpy629/2 (accessed on 8 December 2022).

**Acknowledgments:** In this study, we received useful comments from Junko Furukawa of the University of the Sacred Heart.

**Conflicts of Interest:** The author declares no conflict of interest.

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
