# Peer review of "Normative Corporate Income Tax with Rent for SDGs’ Funding: Case of the U.S."

_sustainability, doi:10.3390/su15043176_

Round 1
Reviewer 1 Report
This is a very interesting topic and the research approach is characterized by originality. Still, I believe that the manuscript would be benefited very much by:
1. A further strengthening of the hypotheses applied on each and every step of the research process, accompanied by a critical discussion of :
a) the limitations and the potential of each hypothesis
b) the use of this hypothesis on earlier relative research attempts on this topic (if any)
2. A detailed clarification of the suitability of using Shimamoto (2018) approach for analysis of a) relevant data of other countries and b) Japanese data from 2014 onwards
3. A critical presentation of previous efforts to address the topic, using alternative approaches, as compared to the one suggested by the manuscript.
Reviewer 2 Report
The pure mathematical model presented in this paper (section 2) might be of interest to a journal of microeconomics theory or industrial organization. However, with regard to issues of public finance and, specifically, corporate income tax, this work lacks a clear theoretical framework and research strategy. The author offers a mixture of various issues and research matters which are relevant to public finance and public choice, but this research fails to focus on issues of corporate income tax. Different problems of taxation and public economics are merely juxtaposed without a thorough presentation or discussion of the links between those issues and the analyses presented here. In particular, there seems to be no clear understanding of real-world issues of corporate income tax. The attempts to provide some justification for the kind of analyses reported in this paper are very weak. I am afraid this paper does not offer any scientific contribution as to how corporate income tax should be changed (and improved).

Reviewer 3 Report
The comments are in the document within the attachment.

Round 2
Reviewer 2 Report
I do appreciate the great amount of work that was done by the author to revise this manuscript in a short time. Many parts of the previous draft that were unnecessary or spurious were deleted; some parts were changed, and there are also new sections in the latest draft. However, my major concern is still about the scientific contribution of this work.
Many of the comments about the previous draft were addressed by simply deleting text, instead of solving those issues. Therefore, there are additional weaknesses from this.
A core piece of this paper is the model presented in section 2, which was originally published by Shimamoto in 2018, and that is applied in this paper “with a few changes” (line 433, as added after revisions). Those changes are not specified in detail and it is not easy to appreciate the actual contribution beyond Shimamoto (2018). In addition, the model considers both product and factor markets, whereas the current version of this work seems, from the introduction, to focus on monopsony only (according to the author’s response and subsequent changes to the original draft, especially as regards the introduction).
The results presented in section 3 are calculated using data from 234 corporations in the US S&P 500. The current draft includes the names of the 28 industries considered in this part of the paper. However, it is not clear why those industries are considered in this research context, as the focus should be on (monopsony) rents. The existence of rents in all of those industries seems to be an unrealistic assumption, as only a few of those industries might show elements of monopsony (or monopoly).
This research appears to assume that corporations enjoy rents and that they should be taxed. They are not taxed as much as they should because they lobby politicians. Then, that taxes can be raised without incurring limits (one of the major issues raised in my previous reports), and that revenues should be used by governments for SDGs funding. Of course, I am oversimplifying here. But, as highlighted earlier, public finance issues are much more complex than assumed by the author and the paper does not offer guidance in order to appreciate the deep interconnections among the many aspects that are discussed in the introduction (which is too parsimonious in providing references to relevant literature).
Although issues of public economics are presented in somewhat clearer and more rigorous ways in the current draft, this research continues to lack solid economic bases. The introduction, perhaps, is still too long and vague because of this. Turning to section 4.1 (Key findings and policy recommendations), vagueness and lack of solid knowledge of public finance is apparent again from lines 724-736.
Section 1.1.2 should have a better focus on issues of taxation.
A couple of minor comments, keeping this part of my report very very short:
1) the first line of the abstract (line 8) states something that the author does not do, as the current argument is (merely) that taxes should be used to finance SDGs; instead, lines 55-56 provide a better description of this research;
2) Section 1.1.1 introduces ecological economics into the discussion; it also refers to the use of production inputs mainly by manufacturing producers. However, the 28 industries considered include much more than manufacturing producers. The paper presents these kinds of weaknesses very often, which contribute to an increasing lack of focus and clarity;
3) Lines 155-157 propose a relevant statement for the analysis carried out by the author, but there is no reference to the literature nor a solid and thorough argument from the author;
4) Lines 241-242 and 242-244 look the same and this should obviously be amended.
Round 3
Reviewer 2 Report
Paper is OK.